# FSV: Learning to Factorize Soft Value Function for Cooperative Multi-Agent Reinforcement Learning

## Abstract

We explore stochastic-based policy solutions for cooperative multi-agent reinforcement learning (MARL) using the idea of function factorization in centralized training with decentralized execution (CTDE). Existing CTDE based factorization methods are susceptible to the relative overgeneralization, where finding a suboptimal Nash Equilibrium, which is a well-known game-theoretic pathology. To resolve this issue, we propose a novel factorization method for cooperative MARL, named FSV, which learns to factorize the joint soft value function into individual ones for decentralized execution. Theoretical analysis shows that FSV solves a rich class of factorization tasks. Our experiments for the well-known tasks of the Non-Monotonic Matrix game and the Max of Two Quadratics game show that FSV converges to optima in the joint action space in the discrete and continuous tasks by local searching. We evaluate FSV on a challenging set of StarCraft II micromanagement tasks, and show that FSV significantly outperforms existing factorization multi-agent reinforcement learning methods.

## 1 Introduction

Cooperative multi-agent reinforcement learning (MARL) aims to instill in agents policies that maximize the team reward accumulated over time (Panait & Luke (2005); Busoniu et al. (2008); Tuyls & Weiss (2012)), which has great potential to address complex real-world problems, such as coordinating autonomous cars (Cao et al. (2013)). Considering the measurement and communication limitations in practical problems, cooperative MARL faces the partial observability challenge. That is, each agent chooses actions just based on its local observations.

Centralized training with decentralized execution (CTDE) (Oliehoek et al. (2011)) is a common paradigm to address the partial observability, where agents' policies are trained with access to global information in a centralized way and executed only based on local observations in a decentralized way, such as the MADDPG (Lowe (2017)) and COMA (Foerster et al. (2017)). However, the size of the joint state-action space of the centralized value function grows exponentially as the number of agents increases, which is known as the scalibility challenge.

Value function factorization methods have been an increasingly popular paradigm for solving the scalability in CTDE by satisfying the Individual-Global-Max (IGM) where the optimal joint action selection should be consistent with the optimal individual action selections. Three representative examples of value function factorization methods include VDN (Sunehag et al. (2017)), QMIX (Rashid et al. (2018)), and QTRAN (Son et al. (2019)). All these methods are $\epsilon$-greedy policies, where VDN and QMIX give sufficient but unnecessary conditions for IGM by additivity and monotonicity structures respectively, and the QTRAN formulates the IGM as an optimization problem with linear constraints.

Although these methods have witnessed some success in some tasks, they all face relative overgeneralization, where agents may stick into a suboptimal Nash Equilibrium. In fact, relative overgeneralization is a grave pathology arising which occurs when a suboptimal Nash Equilibrium in the joint space of action priors to an optimal Nash Equilibrium since each agent's action in the suboptimal equilibrium is a better choice (Wei & Luke (2016)). The non-monotonic matrix game is a simple discrete example. Both VDN and QMIX fail to learn the optimal policy in the non-monotonic

matrix due to their structure limitation. Although QTRAN expresses the complete value function representation ability in the non-monotonic matrix, its full expressive ability decreases in the complex tasks due to the computationally intractable constraints relaxing with tractable L2 penalties. Besides, QTRAN sacrifices the tractability in continuous action space. Therefore, in discrete and continuous tasks, achieving effective scalability while avoiding relative overgeneralization remains an open problem for cooperative MARL.

To address this challenge, this paper presents a new definition of factorizable tasks called IGO (*Individual-Global-Optimal*) which introduces the consistency of joint optimal stochastic policies and individual optimal stochastic policies. Theoretical analysis shows that IGO degenerates into IGM if the policy is greedy, which represents the generality of IGO. Under the IGO, this paper proposes a novel factorization solution for MARL, named FSV, which learns to factorize soft value function into individual ones for decentralized execution enabling efficient learning and exploration through maximum entropy reinforcement learning. To our best knowledge, FSV is the first multi-agent algorithm with stochastic policies using the idea of factorization, and theoretical analysis shows that FSV solves a rich class of tasks.

We evaluate the performance of FSV in both discrete and continuous problems proposed by Son et al. (2019); Wei et al. (2018) and a range of unit micromanagement benchmark tasks in StarCraft II. The Non-Monotonic Matrix game shows that FSV has full expression ability in the discrete task, and the Max of Two Quadratics game shows that FSV is the first factorization algorithm that avoids the relative overgeneralization to converge to optima in the continuous task. On more challenging StarCraft II tasks, due to the high representation ability and exploration efficiency of FSV, it significantly outperforms other baselines, SMAC (Samvelyan et al. (2019)).

## 2 PRELIMINARIES

### 2.1 DEC-POMDP AND CTDE

A fully cooperative multi-agent task can be described as a Dec-POMDP defined by a tuple $\mathcal{G} = \langle \mathcal{S}, \mathcal{U}, \mathcal{P}, r, \mathcal{Z}, \mathcal{O}, \mathcal{N}, \gamma \rangle$, where $s \in \mathcal{S}$ is the global state of the environment. Each agent $i \in \mathcal{N}$ choose an action $u_i \in \mathcal{U}$ at each time step, forming a joint action $u \in \mathcal{U}^{\mathcal{N}}$. This causes a transition to the next state according to the state transition function $\mathcal{P}(s'|s, u) : \mathcal{S} \times \mathcal{U}^{\mathcal{N}} \times \mathcal{S} \to [0, 1]$ and reward function $r(s, u) : \mathcal{S} \times \mathcal{U}^{\mathcal{N}} \to \mathcal{R}$ shared by all agents. $\gamma \in [0, 1]$ is a discount factor. Each agent has individual, partial observation $z \in \mathcal{Z}$ according to observation function $\mathcal{O}(s, i) : \mathcal{S} \times \mathcal{N} \to \mathcal{Z}$. Each agent also has an action-observation history $\tau_i \in \mathcal{T} : (\mathcal{Z} \times \mathcal{U})^*$, on which it conditions a stochastic policy $\pi_i(u_i|\tau_i) : \mathcal{T} \times U \to [0, 1]$. The joint policy $\pi$ has a joint action-value function $Q^{\pi}(s_t, u_t) = \mathcal{E}_{s_{t+1:\infty}, u_{t+1:\infty}}[\sum_{k=0}^{\infty} \gamma^k r_{t+k}|s_t, u_t]$.

Centralized Training with Decentralized Execution (CTDE) is a common paradigm of cooperative MARL tasks. Through centralized training, the action-observation histories of all agents and the full state can be made accessible to all agents. This allows agents to learn and construct individual action-value functions correctly while selecting actions based on its own local action-observation history at execution time .

### 2.2 VDN, QMIX AND QTRAN

An important concept for factorizable tasks is IGM which asserts that the joint action-value function $Q_{tot} : \mathcal{T}^N \times U^N \to R$ and individual action-value functions $[Q_i : \mathcal{T} \times U \to R]_{i=1}^N$ satisfies

$$\arg\max_u Q_{tot}(\tau, u) = (\arg\max_{u_1} Q_1(\tau_1, u_1), ..., \arg\max_{u_N} Q_N(\tau_N, u_N)) \tag{1}$$

To this end, VDN and QMIX give sufficient conditions for the IGM by additivity and monotonicity structures, respectively, as following:

$$Q_{tot}(\tau, u) = \sum_{i=1}^N Q_i(\tau_i, u_i) \quad \text{and} \quad \frac{\partial Q_{tot}(\tau, u)}{\partial Q_i(\tau_i, u_i)} > 0, \forall i \in N \tag{2}$$

However, there exist tasks whose joint action-value functions do not meet the said conditions, where VDN and QMIX fail to construct individual action-value function correctly. QTRAN uses a linear constraint between individual and joint action values to guarantee the optimal decentralisation.

To avoid the intractability, QTRAN relax these constraints using two L2 penalties. However, this relaxation may violate the IGM and it has poor performance on multiple multi-agent cooperative benchmarks as reported recently.

### 2.3 THE RELATIVE OVERGENERALIZATION PROBLEM

Relative overgeneralization occurs when a sub-optimal Nash Equilibrium (e.g. $N$ in Fig. 1) in joint action space is preferred over an optimal Nash Equilibrium (e.g. $M$ in Fig. 1) because each agent's action in the suboptimal equilibrium is a better choice when matched with arbitrary actions from the collaborating agents. Specifically, as shown in Figure 1, where two agents with one-dimensional bounded action (or three actions in discrete action space) try to cooperate and find the optimal joint action, the action $B$ (or $C$) is often preferred by most algorithms as mentioned in (Son et al. (2019) and Wei et al. (2018)) due to their structure limitation and lack of exploration.

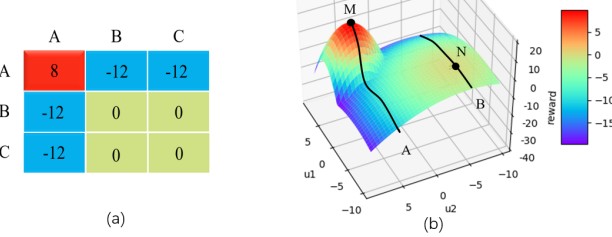

(a)  (b)

Figure 1: The relative overgeneralization in discrete (a) and continuous (b) action space

## 3 METHOD

In this section, we will first introduce the IGO (*Individual-Global-Optimal*), a new definition of factorizable MARL tasks with stochastic policies. Theoretical analysis shows that IGO degenerates into IGM if the policy is greedy. With the energy-based policy, the structure between joint and individual action values of IGO can be explicitly constructed, which is a novel factorization stochastic-based policy solution we proposed, named FSV. Specifically, FSV realizes IGO using an efficient linear structure and learns stochastic policies through maximum entropy reinforcement learning.

### 3.1 INDIVIDUAL GLOBAL OPTIMAL

In the CTDE paradigm, each agent $i \in N$ chooses an action based on a stochastic policy $\pi_i(u_i|\tau_i)$ at the same time step. The joint policy $\pi_{tot}(u|\tau) = \prod_{i=1}^{N} \pi_i(u_i|\tau_i)$ describes the probability of taking joint actions $u$ on joint observation history $\tau$. If each agent adopts its optimal policy while the joint policy is exactly the optimum, the task itself can achieve global optimum through local optimum, which naturally motivates us to consider the factorizable tasks with stochastic policy as following:

**Definition 1** *For a joint optimal policy $\pi_{tot}^*(u|\tau) : \mathcal{T}^N \times U^N \to [0, 1]$, if there exists individual optimal policies $[\pi_i^*(u_i|\tau_i) : \mathcal{T} \times U \to [0, 1]]_{i=1}^{N}$, such that the following holds*

$$\pi_{tot}^*(u|\tau) = \prod_{i=1}^{N} \pi_i^*(u_i|\tau_i) \tag{3}$$

*then, we say that $[\pi_i]$ satisfy IGO for $\pi_{tot}$*

As specified above, IGO requires the consistency of joint optimal policy and individual optimal policies rather than the actions in IGM, but it degenerates into IGM if policies are greedy. That is to say, IGO is more generality than IGM.

### 3.2 FSV

In this work, we take the energy-based policies as joint and individual optimal policy respectively,

$$\pi_{tot}^*(u|\tau) = \exp(\frac{1}{\alpha}(Q_{tot}(\tau, u) - V_{tot}(\tau))) \tag{4}$$

$$\pi_i^*(u_i|\tau_i) = \exp(\frac{1}{\alpha_i}(Q_i(\tau_i, u_i) - V_i(\tau_i))) \tag{5}$$

where $\alpha, \alpha_i$ are temperature parameters, $V_{tot}(\tau) = \alpha \log \int_{U^N} \exp(\frac{1}{\alpha} Q_{tot}(\tau, u)) du$ and $V_i(\tau_i) = \alpha_i \log \int_U exp(\frac{1}{\alpha_i} Q_i(\tau_i, u)) du$ are partition functions.

The benefit of using energy-based policy is that it is a very general class of distributions that can represent complex, multi-modal behaviors Haarnoja et al. (2017). Moreover, energy-based policies can easily degenerate into greedy policies as $\alpha, \alpha_i$ anneals.

To learn this decentralized energy-based policy, we extend the maximum entropy reinforcement learning framework for the multi-agent setting, which we'll describe in the next. Another benefit of considering the stochastic policy with explicit function class for factorizable tasks through IGO is that the architecture between joint and individual action values can be easily constructed through its constrains on policies with specific meanings as follows.

**Theorem 1** *If the task satisfies IGO, with energy-based optimal policy, the joint action value $Q_{tot}$ can be factorized by individual action values $[Q_i]_{i=1}^N$ as following:*

$$Q_{tot}(\tau, u) = \sum_{i=1}^N \lambda_i^* [Q_i(\tau_i, u_i) - V_i(\tau_i)] + V_{tot}(\tau) \tag{6}$$

*where $\lambda_i^* = \alpha/\alpha_i$.*

Theorem 1 gives the decomposition structure like VDN—the joint value is a linear combination of individual values weighted by $\lambda_i^* > 0$. However, the function class defined by Eq(6), which should only concern the task itself, is related to and limited by the distributions of policy. Although energy-based distribution is very general which has the representation ability of most tasks, to establish the correct architecture between joint and individual Q-values and enable stable learning, we need to extend the function class into any distributions. The key idea is that we approximate the weight vector $\lambda_i$ directly as $\alpha, \alpha_i$ is zero instead of annealing $\alpha_i$ during training process. This extends the function class and will at least guarantee IGM constraint when $\alpha, \alpha_i$ is zero .

**Theorem 2** *When $\alpha, \alpha_i \to 0$, the function class defined by IGM is equivalent to the following*

$$Q_{tot}(\tau, u) = \sum_{i=1}^N \lambda_i(\tau, u)[Q_i(\tau_i, u_i) - V_i(\tau_i)] + V_{tot}(\tau) \tag{7}$$

*where $\lambda_i(\tau, u) = \lim_{\alpha, \alpha_i \to 0} \lambda_i^*$.*

Note that $\lambda_i$ is now a function of observations and actions due to the relaxation. Eq(7) allows us to use a simple linear structure to train joint and individual action values efficiently and guarantee the correct estimation of optimal Q-values. We'll describe it in experiment.

Then, we introduce the maximum entropy reinforcement learning in CTDE setting which is an directly extension of soft actor-critic (q-learning).

The standard reinforcement learning tries to maximum the expected return $\sum_t E_\pi[r_t]$, while the maximum entropy objective generalizes the standard objective by augmenting it with an entropy term, such that the optimal policy additionally aims to maximize its entropy at each visited state

$$\pi_{MaxEnt} = \arg\max_\pi \sum_t E_\pi[r_t + \alpha H(\pi(\cdot|s_t))] \tag{8}$$

where $\alpha$ is the temperature parameter that determines the relative importance of the entropy term versus the reward, and thus controls the stochasticity of the optimal policy (Haarnoja et al. (2017)). We can extend it into cooperative multi-agent tasks by directly considering the joint policy $\pi_{tot}(u|\tau)$ and defining the soft joint action-value function as following:

$$Q_{tot}(\tau_t, u_t) = r(\tau_t, u_t) + E_{\tau_{t+1}, ...}[\sum_{k=1}^\infty \gamma^k(r_{t+k} + \alpha H(\pi_{tot}^*(\cdot|\tau_{t+k}))] \tag{9}$$

then the joint optimal policy for Eq(8) is given by Eq(4) (Haarnoja et al. (2017)). Note that we don't start considering decentralized policies, the joint Q-function should satisfy the soft Bellman equation:

$$Q_{tot}^*(\tau_t, u_t) = r_t + E_{\tau_{t+1}}[V_{tot}^*(\tau_{t+1})] \tag{10}$$

And we can update the joint Q functions in centralized training through soft Q-iteration:

$$Q_{tot}(\tau_t, u_t) \leftarrow r_t + E_{\tau_{t+1}}[V_{tot}(\tau_{t+1})] \tag{11}$$

It's natural to take the similar energy-based distribution as individual optimal policies $\pi_i^*$ in Eq(5) which allows us to update the individual policies through soft policy-iteration:

$$\pi_i^{new} = \arg\min_{\pi' \in \prod} D_{KL}(\pi'(\cdot|\tau)||\pi_i^*(\cdot|\tau)) \tag{12}$$

### 3.3 ARCHITECTURE

In this section, we present a novel MARL framework named FSV, which incorporates the idea in a simple and efficient architecture through Eq(7) with multi-agent maximum entropy reinforcement learning. FSV can be applied both in continuous action space and also in discrete action space as a simplification.

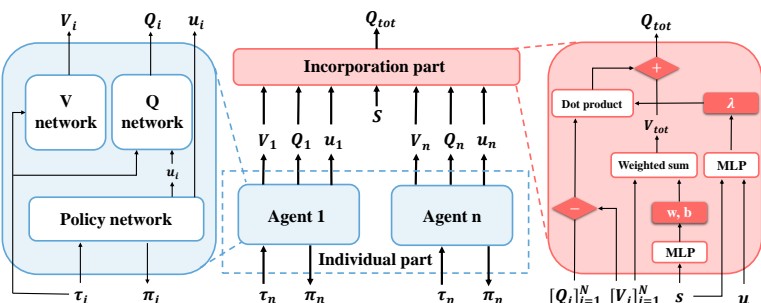

Figure 2: FSV network architecture

Figure 2 shows the overall learning framework, which consists of two parts:(i) individual parts for each agent i, which represents $Q_i$, $V_i$ and $\pi_i$ (ii)incorporation part that composes $Q_i$, $V_i$ to $Q_{tot}$.

Individual parts for each agent i has three networks: (i)individual Q network takes its own action and observation history $\tau_i, u_i$ as input and produces action-values $Q_i(\tau_i, u_i)$ as output.(ii)individual value network takes its own observation history $\tau_i$ as input and produces $V_i(\tau_i)$ as output.(iii)individual policy network takes its own observation history $\tau_i$ as input and produces a distribution (e.g. mean and standard deviation of Gaussian distribution) for sample actions.

Incorporation part composes $Q_i$, $V_i$ to $Q_{tot}$ through linear combination. Specifically, it sums up $[Q_i - V_i]_{i=1}^N$ with coefficients $\lambda_i$ and uses a one-layer hyper-network to efficiently approximate the high-dimensional partition function as following:

$$V_{tot}(\tau) = \sum_{i=1}^N w_i(\tau)V_i(\tau_i) + b(\tau) \tag{13}$$

where $w_i, b$ is a positive weight and bias respectively. To enable efficient learning, we adopt a multi-head attention structure to estimate the weight vector:

$$\lambda_i(\tau, u) = \sum_{h=1}^H \lambda_{i,h}(\tau, u) \tag{14}$$

where $H$ is the number of attention heads and $\lambda_{i,h}$ is defined by

$$\lambda_{i,h} \propto \exp(e_u^T W_{k,h}^T W_{q,h} e_s) \tag{15}$$

where $e_u$ and $e_s$ is obtained by two-layer embedding transformation for $u$ and $s$. The joint action value function $Q_{tot}$ is updated through soft Q-iteration:

$$J_{Q_{tot}}^\theta = E_{(\tau_t, u_t) \sim D}[Q_{tot}(\tau_t, u_t) - \hat{Q}(\tau_t, u_t)]^2 \tag{16}$$

where $\hat{Q}(\tau_t, u_t) = r(\tau_t, u_t) + \gamma E_{\tau_{t+1} \sim D, u_{t+1} \sim \pi}[Q_{tot}(\tau_{t+1}, u_{t+1}) - \alpha \log \pi_{tot}(u_{t+1}|\tau_{t+1})]$.

The individual value network is trained by minimize

$$J_{V_i}^{\phi_i} = E_{\tau_i \sim D}[V_i(\tau_i) - (E_{u_i}[Q_i(\tau_i, u_i) - \alpha \log \pi_i(u_i|\tau_i)])]^2 \tag{17}$$

The policy network of each agent is trained by minimizing the expected KL-divergence

$$J_{\pi_i}^{\psi_i} = E_{\tau_i \sim D, u_i \sim \pi_i}[\alpha \log \pi_i(u_i|\tau_i) - Q_i(\tau_i, u_i)] \tag{18}$$

For discrete action space, it's convenient to simplify this framework to Q-learning. Specifically, we directly compute the individual value function $V_i = \alpha_i \log \sum \exp(\frac{1}{\alpha_i} Q_i(\tau_i, \cdot))$ instead of updating the value network, and action distributions are directly produced by Eq(5) instead of the policy network.

## 4 RELATED WORK

There are many early works with maximum entropy principle such as Todorov (2010) and Levine & Koltun (2013) use it in policy search in linear dynamics and Kappen (2005) and A. Theodorou et al. (2010) use it in path integral control in general dynamics. Recent off policy methods (Haarnoja et al. (2017);Haarnoja et al. (2018b);Haarnoja et al. (2018a)) have been proposed to learn an energy-based policy efficiently through the maximum entropy objective which is adopted in our framework. Value function factorization methods start from VDN (Sunehag et al. (2017)), and is extended by QMIX (Rashid et al. (2018)) and QTRAN (Son et al. (2019)). Other methods such as QATTEN (Yang et al. (2020)) and MAVEN (Mahajan et al. (2019)) go a step further on architecture and exploration. Our method are a member of them but out of the deterministic policy

Current methods adopt different ideas to solve the relative overgeneralization problem. Wei et al. (2018) conduct multi-agent soft Q learning for better exploration. Wen et al. (2019) uses probabilistic recursive reasoning to model the opponents, Yu et al. (2019) adopts inverse reinforcement learning to avoid this problem through right demonstrations, Tian et al. (2019) derives a variational lower bound of the likelihood of achieving the optimality for modeling the opponents. However, none of them adopt value function factorization like FSV which means they suffer the scalability problem.

## 5 EXPERIMENTS

In this section, we first consider two simple examples proposed by prior work (Son et al. (2019),Wei et al. (2018)) to demonstrate the optimality and convergence of FSV in discrete and continuous action space respectively. And we evaluate the performance in a challenging set of cooperative StarCraft II maps from the SMAC benchmark (Samvelyan et al. (2019)).

### 5.1 MATRIX GAME

The matrix game is proposed by QTRAN Son et al. (2019), where two agents with three actions and shared reward as illustrated in Table1, should learn to cooperate to find the optimal joint action $(A, A)$. This is a simple example of the relative overgeneralization problem, where the sub-optimal action $B, C$ has higher expected return in exploration process. We train all algorithms through a full exploration (i.e.,$\epsilon = 1$ in $\epsilon$-greedy) conducted over 20,000 steps while FSV is trained by annealing $\alpha$ from 1 to $\alpha_0$. To demonstrate the expressive ability related to temperature parameter $\alpha$, we set $\alpha_0 = 1, 0.1, 0.01$ respectively. As shown in Table3, QMIX fails to represent the optimal joint action

Table 1: Payoff of matrix game

| $u_1$ \ $u_2$ | A | B | C |
|---|---|---|---|
| A | 8 | -12 | -12 |
| B | -12 | 0 | 0 |
| C | -12 | 0 | 0 |

Table 2: QTRAN-alt

| $Q_1$ \ $Q_2$ | 3.3 | 0.1 | 0.1 |
|---|---|---|---|
| 4.7 | 8.0 | -12.0 | -12.0 |
| -0.1 | -12.0 | 0.0 | 0.0 |
| -0.1 | -12.0 | 0.0 | 0.0 |

Table 3: QMIX

| $Q_1$ \ $Q_2$ | -5.6 | 0.1 | 0.1 |
|---|---|---|---|
| -6.6 | -8.1 | -8.1 | -8.1 |
| 0.2 | -8.1 | 0.0 | 0.0 |
| 0.1 | -8.1 | 0.0 | 0.0 |

Table 4: FSV, $\alpha_0 = 0.01$

| $Q_1$ \ $Q_2$ | 3.3 | -0.7 | -0.0 |
|---|---|---|---|
| 4.7 | 8.0 | -12.0 | -12.0 |
| 0.7 | -12.0 | 0.0 | 0.0 |
| 0.7 | -12.0 | 0.0 | 0.0 |

Table 5: FSV, $\alpha_0 = 0.1$

| $Q_1$ \ $Q_2$ | 7.2 | -0.5 | -0.5 |
|---|---|---|---|
| 0.9 | 8.0 | -11.8 | -11.7 |
| 0.5 | -11.8 | -0.0 | 0.0 |
| 0.5 | -11.8 | -0.0 | 0.0 |

Table 6: FSV, $\alpha_0 = 1$

| $Q_1$ \ $Q_2$ | 0.3 | 0.1 | 0.1 |
|---|---|---|---|
| 7.7 | 8.0 | -9.4 | -4.9 |
| 0.0 | 3.8 | 0.1 | -0.0 |
| -0.1 | 3.8 | -0.1 | -0.0 |

value and the optimal action due to the limitation of additivity and monotonicity structures while FSV and QTRAN successfully represent all the joint action values. In addition, even if $\alpha$ is not annealed to very small, FSV correctly approximated the optimal joint action values because we directly estimate $\lambda$ when $\alpha$ and $\alpha_i$ tend to $0$, which relaxes the constraints of the function class to guarantee the correct structure during the training process.

### 5.2 MAX OF TWO QUADRATICS GAME

We use The Max of Two Quadractics game (Wei et al. (2018)), which is a simple single state continuous game for two agents, to demonstrate the performance of current algorithms in the relative overgeneralization problem. Each agent has one dimensional bounded action with shared reward as following

$$\begin{cases} f_1 = h_1 \times [-(\frac{u_1-x_1}{s_1})^2 - (\frac{u_2-y_1}{s_1})^2] \\ f_2 = h_2 \times [-(\frac{u_1-x_2}{s_2})^2 - (\frac{u_2-y_2}{s_2})^2] + c \\ r(u_1, u_2) = \max(f_1, f_2) \end{cases} \quad (19)$$

where $u_1, u_2$ are the actions from agent 1 and agent 2 respectively, $h_1 = 0.8$, $h_2 = 1$, $s_1 = 3$, $s_2 - 1$, $x_1 = -5$, $x_2 = 5$, $y_1 = -5$, $y_2 = 5$, $c = 10$. The reward function is shown as Fig 3(a). Although this game is very simple, the gradient points to the sub-optimal solution at $(x_1, y_1)$ over almost all the action space which will fox the policy-based method. And for value function factorization methods, this task requires non-monotonic structures to correctly represent the optimal joint Q-values through individual Q values. We extend QMIX and VDN to actor-critic framework (like DDPG) while QTRAN is not applicable in continuous action space due to its requirement of $\max$ operations on Q-values.

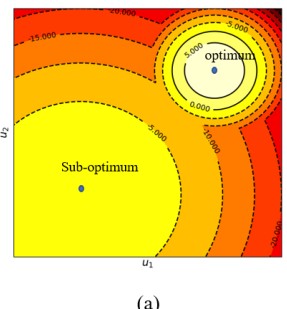

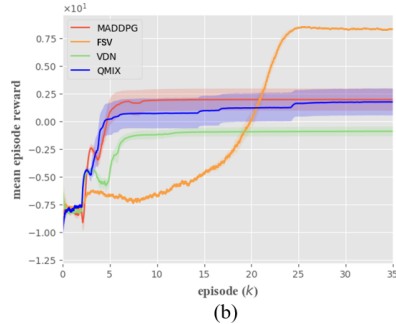

(a)  (b)

Figure 3: Max of Two Quadratics game:(a)reward function, (b)average reward for FSV,VDN,QMIX and MADDPG

Fig 3(b) is the training result averaged over 20 experiment runs and Table 7 gives a more detailed result, where MADDPG and QMIX happened to find the optimal actions due to random initialization twice. VDN never find the optimal actions and even fails to find the sub-optimal 4 times. These

Table 7: training result for Max of Two Quadratics game

|        | opt | sub-opt | other |
|--------|-----|---------|-------|
| FSV    | 20  | 0       | 0     |
| MADDPG | 2   | 18      | 0     |
| QMIX   | 2   | 18      | 0     |
| VDN    | 0   | 16      | 4     |

results indicate that, a more explorative policy and correct estimation of Q-values are both needed to overcome the relative overgeneralization problem. Using a centralized critic like MADDPG to guide the decentralized actors will mislead the policy gradients because it averages the Q-values based on others' policies (**?**). Using individual Q-values to guide actors requires the full expressive ability of factorizable tasks where QMIX and VDN fail to estimate individual Q-values correctly due to the structural limitation as shown in Sec5.1 and QTRAN losts its tractability for continuous tasks. To enable better exploration in joint action space, Wei et al. (2018) adopt multi-agent soft q-learning to avoid the relative overgeneralization problem, but it still uses a centralized critic which suffers scalability and it's very sensitive to how the temperature parameter anneals. It's clear that, FSV utilizes value function factorization method to get correct estimation of individual Q-values and carries exploration with a more explorative energy-based policy can achieve $100\%$ success rate.

## 5.3 STARCRAFT II

We choose a challenging set of cooperative StarCraft II maps from the SMAC benchmark (Samvelyan et al. (2019)). Our evaluation procedure is similar to Samvelyan et al. (2019), where the training process is paused every 100000 time steps to run 32 evaluation episodes with decentralised greedy action selection. We compare FSV with VDN, QMIX and QTRAN on several SMAC maps. Here we present the results for Easy map $2s3z$, Hard map $3s\_vs\_5z$ and Super Hard map $MMM2$, which is classified by Samvelyan et al. Fig 4 shows the test win rate averaged over 5 experiment runs for the different algorithms on the maps. FSV achieves state-of-the-art due to the high representation ability and exploration efficiency

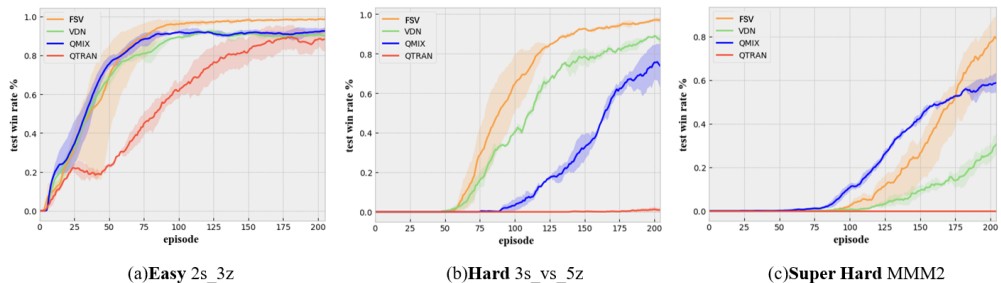

(a)**Easy** 2s_3z       (b)**Hard** 3s_vs_5z      (c)**Super Hard** MMM2

Figure 4: test win rate of FSV, VDN, QMIX and QTRAN

## 6 CONCLUSION

In this paper, we proposed a new definition of factorizable tasks with stochastic policies named IGO. Then we introduced FSV, a novel MARL algorithm under IGO, which learns to factorize soft value function into individual ones for decentralized execution enabling efficient learning and exploration through maximum entropy reinforcement learning. As immediate future work, we aim to develop a theoretical analysis for FSV as a policy-based method. We would also like to explore the committed exploration like Mahajan et al. (2019) in continuous space due to the miscoordination caused by energy-based policy (Wei & Luke (2016)).

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

## 7 APPENDIX

### 7.1 PROOF

#### 7.1.1 RELATIONSHIP BETWEEN IGO AND IGM

If the joint and individual optimal policies are greedy:

$$\pi_i(u_i|\tau_i) = \begin{cases} 1, & u_i = \arg\max_{u_i} Q_i(\tau_i, u_i) \\ 0, & \text{otherwise} \end{cases} \tag{20}$$

$$\pi(u|\tau) = \begin{cases} 1, & u = \arg\max_{u} Q(\tau, u) \\ 0, & \text{otherwise} \end{cases} \tag{21}$$

Then IGO gives that $u = \arg\max_{u} Q(\tau, u)$ if and only if $u_i = \arg\max_{u_i} Q_i(\tau_i, u_i)$ for any $i$ which is equivalent to IGM.

#### 7.1.2 PROOF OF THEOREM 1

**Theorem 1** If the task satisfies IGO, with energy-based optimal policy, the joint action value $Q_{tot}$ can be factorized by individual action values $[Q_i]_{i=1}^N$ as following:

$$Q_{tot}(\tau, u) = \sum_{i=1}^{N} \lambda_i^* [Q_i(\tau_i, u_i) - V_i(\tau_i)] + V_{tot}(\tau) \tag{22}$$

where $\lambda_i^* = \alpha/\alpha_i$.

Proof. Considering Eq(4) and Eq(5), IGO can be reformulated as:

$$\exp(\frac{1}{\alpha}(Q_{tot}(\tau, u) - V_{tot}(\tau))) = \prod_{i=1}^{N} \exp(\frac{1}{\alpha_i}(Q_i(\tau_i, u_i) - V_i(\tau_i))) \tag{23}$$

This gives:

$$Q_{tot}(\tau, u) = \sum_{i=1}^{N} \frac{\alpha}{\alpha_i}[Q_i(\tau_i, u_i) - V_i(\tau_i)] + V_{tot}(\tau) \tag{24}$$

which is Theorem 1.

### 7.1.3 PROOF OF THEOREM2

***Theorem 2*** When $\alpha, \alpha_i \to 0$, the function class defined by IGM is equivalent to the following

$$Q_{tot}(\tau, u) = \sum_{i=1}^{N} \lambda_i(\tau, u)[Q_i(\tau_i, u_i) - V_i(\tau_i)] + V_{tot}(\tau) \tag{25}$$

where $\lambda_i(\tau, u) = \lim_{\alpha, \alpha_i \to 0} \lambda_i^*$.

Proof. **IGM$\Rightarrow$Eq(7)**:

It's clear that Eq(7) can always hold if $\lambda_i$ is well constructed. Here we give one way to construct $\lambda_i$ which meanwhile explains how we extend function class limited by energy-based policy.

Denote $\pi_i, \pi_{tot}$ to be the current policy and $\pi_i^*, \pi_{tot}^*$ to be the optimal policy. We can always take individual optimal policies during the process of approaching greedy, thus

$$\pi_i(u_i|\tau_i) = \exp(\frac{1}{\alpha_i}(Q_i(\tau_i, u_i) - V_i(\tau_i))) = \begin{cases} 1 - \epsilon, & u_i = \arg\max_{u_i} Q_i(\tau_i, u_i) \\ \epsilon, & \text{otherwise} \end{cases} \tag{26}$$

where $\epsilon$ is a small parameter. Then, the joint policy is given by:

$$\pi_{tot}(u|\tau) = \begin{cases} (1-\epsilon)^n, & u = [u_i]_{i=1}^n \\ 1 - (1-\epsilon)^n, & \text{otherwise} \end{cases} \tag{27}$$

Considering IGM, $u = \arg\max_u Q_{tot}(\tau, u)$. Then $\pi_{tot} = \pi_{tot}^* = \exp(\frac{1}{\alpha}(Q_{tot}(\tau, u) - V_{tot}(\tau)))$ and $\alpha, \alpha_i$ is a function of observations and actions:

$$\alpha_i(\tau_i, u_i) = \frac{Q_i(\tau_i, u_i) - V_i(\tau_i)}{\log(1-\epsilon)}, \quad u_i = \arg\max_{u_i} Q_i(\tau_i, u_i) \tag{28}$$

$$\alpha(\tau, u) = \frac{Q_{tot}(\tau, u) - V_{tot}(\tau)}{n \log(1-\epsilon)}, \quad u = \arg\max_u Q_{tot}(\tau, u) \tag{29}$$

Thus, $\lambda_i$ is given by:

$$\lambda_i(\tau, u) = \frac{Q_{tot}(\tau, u) - V_{tot}(\tau)}{n(Q_i(\tau_i, u_i) - V_i(\tau_i))} \tag{30}$$

This makes the Eq(7) permanent. In particular, if sampled action u is exactly current $\arg\max_u Q_{tot}$, then $Q_{tot} = V_{tot}$ and $Q_i = V_i$ when $\alpha, \alpha_i \to 0$ and $\lambda_i$ can be set to 1. Thus we extend the function class and Eq(7) always holds for any action.

**IGM$\Leftarrow$Eq(7)**:

denote $u_i^* = \arg\max_{u_i} Q_i(\tau_i, u_i)$ and $u^* = \arg\max_u Q(\tau, u)$. Remember

$$V_{tot}(\tau) = \alpha \log \int_{U^N} \exp(\frac{Q_{tot}(\tau, u)}{\alpha}) du \geq Q_{tot}(\tau, u) \tag{31}$$

$$V_i(\tau_i) = \alpha_i \log \int_U \exp(\frac{Q_i(\tau_i, u_i)}{\alpha_i}) du \geq Q_i(\tau_i, u_i) \tag{32}$$

Therefore, $Q_{tot} = V_{tot}$ and $Q_i = V_i$ if and only if $u = u^*$ and $u_i = u_i^*$, respectively. Considering Eq(7) and $\lambda_i > 0$, $u = u^*$ if and only if $u_i = u_i^*$, that complete the proof.

Table 8: hyper-parameters

| settings | discrete | continuous |
|---|---|---|
| FSV | | |
| layer number of $\lambda$ | 2 | 2 |
| unit number of hidden layer in $\lambda$ | 64 | 64 |
| layer number of $w, b$ | 1 | 1 |
| unit number of hidden layer in $w, b$ | $\emptyset$ | 64 |
| layer number of actor | $\emptyset$ | 2 |
| unit number of hidden layer in actor | $\emptyset$ | 64 |
| learning rate of actor | $\emptyset$ | $3e - 4$ |
| $\alpha$ decay scheme | linear decay from 1 to 0.01 | Automate Entropy Adjustment |
| QMIX and VDN | | |
| layer number of actor | $\emptyset$ | 2 |
| unit number of hidden layer in actor | $\emptyset$ | 64 |
| learning rate of actor | $\emptyset$ | 3e-4 |

## 7.2 IMPLEMENTATION DETAILS

In discrete tasks, We follow the PyMARL Samvelyan et al. (2019) implementation of VDN, QMIX and QTRAN, where the hyper-parameters of are the same in SMAC Samvelyan et al. (2019). We illustrate the special hyper-parameters of FSV in Table 8 and others are the default settings in Py-MARL. In discrete tasks, we extend VDN and QMIX to the actor-critic framework. Specifically, we add an actor for individual agents to maximise the Q-values from their own critic like DDPG. We illustrate the hyper-parameters of these algorithm as well as FSV in Table 8. For stability, we reformulate the Eq(7) as following:

$$Q_{tot}(\tau, u) = \sum_{i=1}^{N} \lambda_i(\tau_i, u_i)(Q_i(\tau_i, u_i) - V_i(\tau_i)) + V_{tot}(\tau) - \sum_{i=1}^{N} Q_i(\tau_i, u_i) + \sum_{i=1}^{N} Q_i(\tau_i, u_i) \quad (33)$$

Then we stop all the gradients except for $\lambda_i$ and the last term $Q_i$. Considering incorrect weight $\lambda_i$ will cause incorrect $\alpha_i$ at the beginning of training thus obstruct the exploration, we use $\alpha_i = \alpha$ , which means we ignore the KL-divergence between current policy and optimal policy. The only temperature parameter $\alpha$ is updated through annealing or Automating Entropy Adjustment in Haarnoja et al. (2018b) as following:

$$J_\alpha = E_{u \sim \pi}[-\alpha \log \pi(u|\tau) - \alpha \bar{H}] \quad (34)$$

where the $\bar{H}$ is the target entropy.

