# OpenReview forum: "FSV: Learning to Factorize Soft Value Function for Cooperative Multi-Agent Reinforcement Learning"
_ICLR.cc/2021/Conference — Reject_

### Official Review · AnonReviewer2 · 2020-10-28
**Without citation to an earlier paper, QPLEX [https://arxiv.org/abs/2008.01062], but the factorization and many details are identical.**

**Rating:** 3
**Confidence:** 4

**Review:**

In this paper, the authors use a method in soft RL style and a QPLEX factorization to propose the first multi-agent value function decomposition (VFD) method for stochastic policies. I appreciate the efforts to extend VFD to a larger function class.

However, I am confused about the similarity to QPLEX [https://arxiv.org/abs/2008.01062v1 ], which was published several months ago. One of the main contributions of the paper is the value function factorization stated in Theorem 2. However, it is identical to QPLEX, including notations. Specifically, Eq. 7 and 13 of FSV are the same as Eq. 8, 10, and 12 in QPLEX, including notations.

Moreover, some statements in the paper and details of the figures are very similar to those in the QPLEX paper.  Eq. 33 of FSV is the same as Eq. 50 of QPLEX. These equations describe an implementation detail where QPLEX and FSV stop gradients at the same variables. Stopping gradients is reasonable in the context of QPLEX, and the reason for using this trick is well-motivated there. By contrast, why this trick is necessary for FSV remains largely unclear. I encourage the authors to explain why they stop gradients here in the context of soft reinforcement learning.

In summary, the similarity including theorems, equations, and notations (and even figures). Despite these similarities, the authors did not cite QPLEX, which has been online for several months before FSV is published.

Additionally, about the integration of QPLEX and soft Q-learning, I also have some concerns. The definition of soft value functions depends on the specific selection of the temperature parameter. In this paper, the temperature parameters are end-to-end learned by minimizing the temporal-difference error of Q-learning. Could the authors explain why this makes sense in the framework of soft Q-learning? The counterpart in QPLEX ($\lambda$ in Eq. 10) is end-to-end learned to ensure the rich expressivity of QPLEX so that it can represent the complete IGM function class. This design is well-motivated in the original QPLEX paper.


** Minor points
Some of the claims in the paper need refinements. For example, in Para. 2 of the introduction, "Centralized training with decentralized execution (CTDE) (Oliehoek et al. (2011)) is a common paradigm to address the partial observability". This claim is not solid. Partial observability can still induce miscoordination in decentralized execution [Wang et al. ICLR 2020, https://openreview.net/forum?id=HJx-3grYDB]. That is to say, the framework of CTDE itself cannot solve the problem of partial observability. It is better to say something like "CTDE with communication." I also find the claim "Value function factorization methods have been an increasingly popular paradigm for solving the scalability in CTDE" not convincing. In fact, QMIX, which is a VFD method, generally can not work very well in tasks with more than 20 agents.

Typos: Last paragraph in the introduction: "it significantly outperforms other baselines, SMAC (Samvelyan et al. (2019))." SMAC is not an algorithm.

---

> ### Author Response · Authors · 2020-11-23
> **Response to Reviewer 2**
>
>
> Thanks for your thoughtful comments.
>
> It should be mentioned that we have not copied QPLEX since our work and QPLEX has different motivation and contribution.
> Motivation. QPLEX followed the IGM principle and proposed a duplex dueling network to achieve a complete IGM function class. However, IGM cannot be applied to continuous action space since its argmax operator requires the discreteness of Q-values.
> Thus, QPLEX is not suitable for continuous works.
> To address the continuous tasks, we rethink factorizability in a policy-based manner.
> In addition, considering that [1] claimed that overcoming relative overgeneralization required a more explorative approach than simple epsilon-greedy action selection, we introduced a novel factorization algorithm based on the soft policy, called FSV.
> In summary, QPLEX considers the Q value decomposition under the IGM architecture. Its policy is epsilon-greedy, which is suitable for discrete tasks. Our proposed FSV considers the soft policy factorization under the IGO architecture which is not only suitable for discrete but also for continuous tasks.
> Therefore, our work and QPLEX has different motivation and contribution.
>
> Although some details of our work look similar to QPLEX, they are actually different.
> Equation 7 in FSV (FSV:equ7) seems similar to equation 10 in QPLEX (QPLEX:equ10). However, the meanings of Q and V are different in the two papers. The Q in FSV represents the expected return with entropy term, and the V in FSV is the log partition function of Boltzmann distribution, while the Q and V in QPLEX come from the dueling Q structure. In addition, FSV deduced FSV:equ7 directly from the integration of IGO and soft policy, while QPLEX devised QPLEX:equ10 to enforce IGM consistency.
> Both our work and QPLEX utilized an end-to-end learning architecture to approximate weight vectors, the weight vectors represent different meanings in the two papers.
> In QPLEX, the weight vector motivation may origin from Qatten [2]. In our work, the weight vector reveals the connection between credit assignment and exploration.
> Specifically, the larger $\lambda_i$ (which means its $\alpha_i$ is smaller and its policy has less randomness) the more important this agent contributes to the team, thus its policy should be more greedy to keep the team return. Agents with smaller $\lambda_i$(which means its $\alpha_i$ is larger and its policy has more randomness) can explore more arbitrarily because their performance doesn’t matter much.
> In fact, such an end-to-end architecture has been widely used in previous works, such as QMIX and Qatten.
>
> Theorem 2 was regarded as one of the main contributions because the counterpart in QPLEX is its main contribution by the reviewer. However, theorem 2 in FSV is a simple corollary of the integration of IGO and soft policy, since soft policy degenerate to the greedy policy when $\alpha=0$ and IGO degenerate to IGM when the policy is greedy. Therefore, it is clear that FSV can achieve IGM in such a special case.
>
> We stop the gradient there because we found such a trick helps when we duplicated Qatten. This trick will not change the gradient of the weight vector, and it only removes the coefficient $\lambda_i$ of $Q_i's$ gradient of backpropagation which helps to reduce variance especially when $\lambda_i$ is not normalized.
>
> [1] Wei E, Luke S. Lenient learning in independent-learner stochastic cooperative games[J]. The Journal of Machine Learning Research, 2016, 17(1): 2914-2955.
>
> [2] Yang Y, Hao J, Liao B, et al. Qatten: A General Framework for Cooperative Multiagent Reinforcement Learning[J]. arXiv preprint arXiv:2002.03939, 2020.

---

### Official Review · AnonReviewer1 · 2020-10-28
**Poor writing with incomplete/inconsistent proofs**

**Rating:** 2
**Confidence:** 5

**Review:**

This paper is concerned with the problem of cooperative multi-agent reinforcement learning for CTDE scenario which is well studied in recent literature. The authors propose a factorisation method based on soft value functions. I found that the paper is extremely poorly written which makes it very difficult to understand the overall method. The presentation is also quite arbitrary with discussion around results that seem unnecessary. There is little novelty as most of the paper borrows from SAC paper by Harnooja et al, albeit with gross errors in copying. Here are some of the  major issues:

1. The authors discuss the IGO decentralisability, however what is the relation between IGO optimal policy and soft policy when the former is not representable by latter?

2. How does the local soft policy iteration guarantee joint policy improvement?

3. Why is the * being arbitrarily switched in sec 3.2? What does eq 12 even imply? isn't the KL minimiser $\pi_i^*$ itself? Where is $\Pi$ defined?

4. How did eq 18 come about?

5. The paper is full of unbacked blanket statements like: " Although energy based distribution is very general which has the representation ability of most tasks," "Our method are a member of them but out of the deterministic policy" etc.

6. There are many unintelligible sentences like: "we need to extend the function class into any distributions", "IGO is more generality than IGM", "The individual value network is trained by minimize", "relative overgeneralization, where finding a
suboptimal Nash Equilibrium, which is a well-known game-theoretic pathology" etc.

7. In proof for Theorem 2, $\epsilon$-greedy eq 26 cannot be matched by a soft policy in general, thus the rest of the proof cant follow without corrections.

---

> ### Author Response · Authors · 2020-11-23
> **Response to Reviewer 1**
>
> Thanks for your comments.
>
> Q1:what is the relation between IGO optimal policy and soft policy when the former is not representable by latter?
>
> IGO is a definition of the factorization of cooperative tasks in the centralized training phase, which describes the consistency between the optimal joint policy and the collection of individual optimal policies of each agent.
> Specifically, considering the decentralized execution, each individual policy is independent, then, the probability of choosing joint action $\boldsymbol{u}$ naturally equals the product of the probabilities of choosing individual action $u_i$.
> That is, $\prod_{i=1}^{N}\pi_i(u_i|\tau_i) = \pi_{tot}(\boldsymbol{u}|[\tau])$, where $\tau_i$ is a local action-observation history of agent $i$, and $[\tau] = [\tau_i]_{i=1}^N$.
>
> However, when all individual policies based on decentralized information are optimal, the joint policy $\pi_{tot}$ may be optimal or not.
> That is to say, decentralized individual optimal policies may achieve global optimization or not.
> Thus, in the centralized training, when there exists $\pi_{tot}^*(\boldsymbol{u}|\boldsymbol{\tau}) = \prod_{i=1}^{N}\pi_i^*(u_i|\tau_i)$, where $\boldsymbol{\tau} \in \mathcal{T}^N$ is a joint action-observation histories, the global optimization of a cooperative task can be achieved by decentralized individual policies.
> In this case, we say the joint policy can be factorized by individual policies or the task itself is factorizable.
> The relationship between IGO and soft policy is like the relationship between IGM and VDN QMIX QTRAN. Considering that IGM in general corresponds to greedy action selection, whose form of argmax can’t be applied with continuous Q-value, we redefine IGO for continuous space from the perspective of arbitrary policy. This means IGO’s optimal policy can be chosen as not only greedy policy (at which we prove IGO collapse into IGM) but also soft policy if we need.
>
> Q2:How does the local soft policy iteration guarantee joint policy improvement?
>
> Considering that $\pi_{tot}=\prod_{i=1}^N\pi_i$ and IGO gives that $\pi_{tot}^*=\prod_{i=1}^N\pi_i^*$, we have
>
> \begin{equation}\nonumber
>     D_{KL}(\pi_{tot}(u|\tau)||\pi_{tot}^*(u|\tau)) =\int\pi_{tot}(u|\tau) log\frac{\pi_{tot}(u|\tau)}{\pi_{tot}^*(u|\tau)}du = \int\prod_{i=1}^N\pi_i(u_i|\tau_i)\sum_{i=1}^Nlog\frac{\pi_i(u_i|\tau_i)}{\pi_i^*(u_i|\tau_i)}du =\sum_{i=1}^N\int\pi_i(u_i|\tau_i)\pi_{-i}(u_{-i}|\tau_{-i})log\frac{\pi_i(u_i|\tau_i)}{\pi_i^*(u_i|\tau_i)}du_idu_{-i} =\sum_{i=1}^N\int\pi_i(u_i|\tau_i)log\frac{\pi_i(u_i|\tau_i)}{\pi_i^*(u_i|\tau_i)}du_i =\sum_{i=1}^ND_{KL}(\pi_{i}(u_i|\tau_i)||\pi_{i}^*(u_i|\tau_i))
> \end{equation}
> where we rewrite $\pi_{tot}=\pi_i\pi_{-i}$ and $du=du_idu_{-i}$. The integral of -i equals 1 due to the probability normalization. Thus, minimizing the KL of individual policies through soft policy iteration will minimize the KL of joint policy. The following proof is the same as in SAC's paper.
>
> Q3:What is the meaning of * in sec 3.2? What does eq12 imply?
>
> The $\pi$ with * represents the optimal policy. The $\pi$ without * represents the actual policy during training. The Q with * represents the ideal Q value. The Q without * represents the actual Q value during training.
> Eq12 represents the soft policy improvement of each agent, where $\Pi$ is some set of policies such as a parameterized family of Gaussian distributions.
>
> Q4:How did eq 18 come about?
>
> Just like Eq12 in SAC, Eq18 is the way we realize soft policy improvement, where we minimize the KL of the current individual policy and optimal individual policy.
>
> Q7.Proof of theorem2.
>
> $\epsilon-greedy$ indeed cannot be matched by a soft policy in general. But it can be matched when $\epsilon=\alpha=0$, which is all the proof need. In fact, eq7 can hold without IGM if $\lambda$ is well constructed by a neural network.

---

### Official Review · AnonReviewer3 · 2020-10-29
**A novel paper which factorizes soft value function with stochastic policies.**

**Rating:** 4
**Confidence:** 3

**Review:**

This paper proposes a novel MARL framework named FSV, which incorporates the idea of energy-based policies and an efficient linear decomposition architecture in the joint action-value function with multi-agent maximum entropy reinforcement learning. Besides, the authors propose the IGO, which extends the IGM in stochastic policy cases. FSV suits in both the discrete and continuous action space scenarios. Experiments conducted on two simple examples with discrete and continuous action settings show that FSV could overcome the relative overgeneralization problem with the proper temperature setting. Furthermore, FSV in the challenging SMAC benchmark outperforms VDN, QMIX, and QTRAN in three scenarios.

Overall, this paper is well-organized and easy to read. The authors present interesting ideas that combine the energy-based policy and maximum entropy reinforcement learning into the centralized Q-value mixing network to obtain a better expression ability than VDN and QMIX and overcome the relative overgeneralization problem.

There are some questions.

Q1: Does the linear decomposition of Qtot of Qi and Vi limit the representation ability of FSV? It seems that FSV cannot represent the non-linear formation of Qtot and Qi.

Q2: In Section 5.1, it is better to show the estimated \lambda_i’s performance compared with the different \alpha_0 settings.

Q3: In Section 5.3, there lacks analysis of FSV and other methods. Especially, the ablation of FSV should be considered. In these three scenarios, which part (soft RL, critic’s structure, or others) contributes to FSV most and improves its performance steadily?

There are some typos.
In section 5.2, “on others’ policies (?).”
In section 5.3, “exploration efficiency” -> “exploration efficiency.”

As pointed out by other reviewers, this paper shares much similarity with QPLEX but without reference and discussions, thus I would be inclined to reduce the score as well.

---

> ### Author Response · Authors · 2020-11-23
> **Response to Reviewer 3**
>
> Thanks for your comments.
>
> It should be mentioned that our work is different from QPLEX. QPLEX considers the Q value decomposition under the IGM architecture. Its policy is epsilon-greedy, which is suitable for discrete tasks. Our proposed FSV considers the soft policy factorization under the IGO architecture which is not only suitable for discrete but also for continuous tasks. Therefore, our work and QPLEX has different motivation and contribution. We will cite QPLEX in our future version. Hope our reply to Reviewer 2 can clear your doubts.
>
> Q1:  Does the linear decomposition of Qtot of Qi and Vi limit the representation ability of FSV?
>
> The nonlinearity can be obtained by the weight vector since we have end-to-end learned it as a function of $\tau$ and $u$ which can be regarded as a function of $Q_i$.
>
> Thank you for providing the comments in Q2 and Q3 that will help us analyze our method. We will systematically carry out ablation experiments.

---

### Official Review · AnonReviewer4 · 2020-10-29
**Q-functions and Policies**

**Rating:** 2
**Confidence:** 4

**Review:**

The paper proposes a Q-factorization method by assuming an energy-based policies model. Q-functions are formulated as soft value functions with the energy parameters, and this adoption renders the function factorization more flexible compared to existing ones. The proposed solution applies to continuous-action tasks, a feat left unconquered by some of the existing methods. Authors exhibit that FSV outperforms others in various environments characterized by local optima.

Strengths:

+ The formulation of Q-functions as soft functions, despite appearing simple, shows some effectiveness in a number of MARL tasks.

+ The network architecture is intuitive.


Major Concerns:

- Neither energy-based policies nor soft value functions is an original contribution of this work. True, the authors do not claim so. But the reviewer is left unsure as to what then the primary contribution of the paper would be.

- The method generalizes IGM to IGO but in doing so, foregoes the simplicity of the IGM condition. The reviewer would then expect to be met with a somewhat strong guarantee, but is instead presented with approximations on \lambda_i. It is not clear from the paper how much insightful value the method has, when its criticism of a previous work (QTRAN) was based on intractability but the FSV method itself still relies on approximations. It would seem as though QTRAN and FSV each chose different paths to approximate different components of an MARL training scheme - the former takes may stronger assumption on the value functions while the latter takes assumptions on the nature of value functions being parametrized by approximated weights.

- The effectiveness of the proposed method is not yet well-accounted for. Issues are raised, but little explanation (or any attempt thereof) is provided. For example, the reviewer would have very much liked to gain an understanding of the relevance between IGO and its ability to alleviate relative overgeneralization. How does taking on greedy policies (which makes IGO collapse into IGM) make MARL agents more prone to overgeneralize with respect to each other? What kinds of findings would the authors present? What evidence could support those findings? The evaluation, while illustrating great performance gaps, needs a careful redesign so as to construct solid grounds for the soft value function factorization under IGO to be "explainably" better than existing works.

- The paper could be better positioned. The Related Works section could be put to better use to clearly distinguish two very different lines of research: value function factorizing MARL works and maximum entropy principle.

- There needs to be some justification about multi-head attention being used to "enable efficient learning" in Section 3.3. The reviewer is left hanging as to why and how such a choice was made.


Minor Concerns:

* A few parts of the paper were difficult to follow. For example, there is an unfinished sentence in Related Works. In Section 2.1, there is an incomplete clause beginning with "the reward function [...] shared by all agents". Under Theorem 1, "any distributions" --> "any distribution". Also, what is meant by "correct architecture" in that same paragraph?

---

> ### Author Response · Authors · 2020-11-23
> **Response to Reviewer 4**
>
> Thanks for your comments.
>
> Our contribution has three parts. First, we propose the definition of decomposable tasks from the perspective of policy rather than of Q-value, called IGO, which extends greedy-policy-based (value-based) IGM to arbitrary policy.
> Second, based on the IGO, we proposed a new factorize method with an energy-based policy. It should be mentioned that it is the first time to factorize soft policies.
> Third, FSV is not only suitable for discrete action space tasks but also for continuous action space tasks. In addition, FSV solved the relative overgeneralization problem of factorizing methods in continuous action space.
>
> In fact, although IGO requires the consistency of policy which seems like requiring the consistency of all actions, it is not a stronger definition than IGM. The requirement comes from the task itself. That is to say, if achieving the best performance in a given task requires greedy joint action selection, IGO is equivalent to IGM. But if achieving the best performance requires a stochastic joint policy, IGO works while IGM doesn't. In other words, IGO doesn't give a stronger constraint on a given task but extends the definition of factorizability in more tasks. To realize IGO, we adopt soft policy and end-to-end learn the weight vector $\lambda$. Considering that soft policy can collapse into greedy policy when $\alpha=0$, we guarantee IGM constraint in this special case.
>
> There are two reasons for the sub-optimal action selection (relative overgeneralization). First, as QTRAN pointed out, QMIX and VDN cannot represent the optimal due to its lack of expressive ability. Second, as MASQL [1] pointed out, relative overgeneralization prevents policy gradient methods from achieving better coordination. Therefore, there are two ways to solve this problem. One is to enhance the expressive ability of the function class such as QTRAN and QPLEX, which can solve this problem in discrete action space.
> Another is adopting a more explorative approach than a simple epsilon-greedy action selection as [2] says. We have shown in the matrix game that expressive ability is the key to solve this problem in discrete action space. However, in continuous tasks, even with a fully centralized critic that has the best expressive ability like MADDPG, it will still suffer relative overgeneralization [1]. MASQL integrated MADDPG and soft policy and improves its performance, which motivates us to adopt a soft policy under IGO constrain. In conclusion, FSV with the full expressive ability and more explorative soft policy is reasonable to have a better performance.
>
> The multi-head attention structure is inspired by Qatten.
>
> [1] Wei, E., Wicke, D., Freelan, D., & Luke, S. (2018). Multiagent soft Q-learning. ArXiv, March.
>
> [2] Wei E, Luke S. Lenient learning in independent-learner stochastic cooperative games[J]. The Journal of Machine Learning Research, 2016, 17(1): 2914-2955.

---

### Official Review · AnonReviewer5 · 2020-11-04
**Blind review**

**Rating:** 3
**Confidence:** 5

**Review:**

This paper describes a new method for learning factored value functions in cooperative multi-agent reinforcement learning. The approach uses energy-based policies to generate this factorization. The method is presented and experiments are given for smaller domains as well as starcraft.

The idea of learning factored value functions is promising for learning separate value functions for each agent that allow them to learn in a centralized manner and execute in a decentralized manner (centralized training and decentralized execution). Several methods have been proposed along these lines, but as the paper points out, they have limitations that makes them perform poorly in some problems.

The proposed approach in this paper has some promising experimental results, but there are questions about the novelty and significance of the method. Furthermore, evaluating these contributions is difficult due to the lack of clear details in the paper.

In particular, the details of the approach itself in 3 are not clear. Starting with Definition 1, it seems like IGO is using an optimal *centralized* policy. Is this what is meant? If so, why is this needed (as opposed to an optimal decentralized policy). It will typically be impossible to achieve a centralized policy with decentralized information. Furthermore, the energy-based policies are defined in 3.2, but 'key' ideas such as approximating the weight vector aren't fully explained making the exact approach hard to determine. Also, it is beneficial that the current theorems and proofs are included, but the lack of sufficient detail makes it hard to parse and evaluate them.

There are also similar max entropy approaches, such as the paper below.

Iqbal, S. & Sha, F.. (2019). Actor-Attention-Critic for Multi-Agent Reinforcement Learning. Proceedings of the 36th International Conference on Machine Learning, in PMLR 97:2961-2970

As well as other factorized methods, such as the papers below (which are admittedly new).

Weighted QMIX: Expanding Monotonic Value Function Factorisation for Deep Multi-Agent Reinforcement Learning. Tabish Rashid, Gregory Farquhar, Bei Peng, Shimon Whiteson. NeurIPS 2020.

de Witt, Christian Schroeder, et al. "Deep Multi-Agent Reinforcement Learning for Decentralized Continuous Cooperative Control." arXiv preprint arXiv:2003.06709 (2020).

The paper should discuss how the proposed method is an improvement over this other work and have a more comprehensive related work section.

The experiments are promising, but the relevant related work is not included and there isn't sufficient detail describing how the methods were run and discussing the results. In terms of comparisons, the paper should also need to compare with non-factored state-of-the-art methods. It is, of course, natural to compare with other factored methods, but what matters is general state-of-the-art performance of the domains.

As noted, the clarity and writing of the paper should be improved. Beyond the examples above, some other instances are below.

- If the reader doesn't already understand the relative overgeneralization problem, Section 2.3 probably isn't sufficient. Figure 1 is helpful, but it should be described in the text to make the issue clear.

- The connection between the overgeneralization problem and factored representations isn't completely clear. Factored representations have problems because they typically cannot represent the optimal value function (or policy). That is a separate issue than getting stuck in a local optimum (which can happen with any type of method).

---

> ### Author Response · Authors · 2020-11-23
> **Response to Reviewer 5**
>
> Thanks for your comments.
>
> IGO is a definition of the factorization of cooperative tasks in the centralized training phase, which describes the consistency between the optimal joint policy and the collection of individual optimal policies of each agent.
> Specifically, considering the decentralized execution, each individual policy is independent, then, the probability of choosing joint action $\boldsymbol{u}$ naturally equals the product of the probabilities of choosing individual action $u_i$.
> That is, $\prod_{i=1}^N \pi_i(u_i|\tau_i) = \pi_{tot}(\boldsymbol{u} | [\tau])$, where $\tau_i$ is a local action-observation history of agent $i$, and $[\tau] = [\tau_i]_{i=1}^N$.
>
> However, when all individual policies based on decentralized information are optimal, the joint policy $\pi_{tot}$ may be optimal or not.
> That is to say, decentralized individual optimal policies may achieve global optimization or not.
> Thus, in the centralized training, when there exists $\pi_{tot}^*(\boldsymbol{u}|\boldsymbol{\tau}) = \prod_{i=1}^{N}\pi_i^*(u_i|\tau_i)$, where $\boldsymbol{\tau} \in \mathcal{T}^N$ is a joint action-observation histories, the global optimization of a cooperative task can be achieved by decentralized individual policies.
> In this case, we say the joint policy can be factorized by individual policies or the task itself is factorizable.
>
> We deduced Equation 6 through the integration of IGO and soft policy, that is, $Q_{tot}(\tau,u)=\sum_{i=1}^N \frac{\alpha}{\alpha_i}[Q_i(\tau_i,u_i) - V_i(\tau_i)] + V_{tot}(\tau)$.
> Naturally, the $Q_i$ and $V_i$ can be approximated by networks.
> Then, we should find a way to approximate $\alpha$ and $\alpha_i$.
> Considering that the weight vector $\frac{\alpha}{\alpha_i}$ involves the credit assignment of each agent, we can use some global information in the centralized training phase to end-to-end learn the weight vector directly.
>
> Previous work [1] showed a great performance gap between factored and no-factored methods in SCII such as MADDPG and MAAC[2], it points out that the factorization is the key to the performance. Thus, we compare the factored methods in SCII.
> In the Max of Two Quadratics game, MASQL[3] which also utilized soft policy can only converge the global optima for 72$\%$ of the time while MADDPG never converged to it.
> The factored methods which are admitted new such as WQMIX[4] MAVEN[5] and QPLEX[6] can’t be applied in continuous action space. [1] and [7] (of which we follow the implementation in our experiments) extended QMIX and VDN to continuous action space, however, they also inherit their shortcomings as shown in Max of Two Quadratics game.
>
> The relative overgeneralization problem represents a phenomenon that a sub-optimal joint action is preferred than the global optimum. This may be due to a lack of expressive ability to represent the optimum or just getting stuck in the sub-optimum. The previous work [8] have shown that giving higher weight to the larger reward helps to overcome relative overgeneralization. This idea is generally the same as MAVEN[5] which visits the global optimum more often and WQMIX[4] which gives a higher weight to the loss of larger $Q_{tot}$. Therefore, we think its beneficial to consider relative overgeneralization problem in factorize method.
>
> [1]de Witt, C. S., Peng, B., Kamienny, P. A., Torr, P., Böhmer, W., & Whiteson, S. (2020). Deep Multi-Agent Reinforcement Learning for Decentralized Continuous Cooperative Control. ArXiv.
>
> [2] Iqbal, S. & Sha, F.. (2019). Actor-Attention-Critic for Multi-Agent Reinforcement Learning. Proceedings of the 36th International Conference on Machine Learning, in PMLR 97:2961-2970
>
> [3] Wei, E., Wicke, D., Freelan, D., & Luke, S. (2018). Multiagent soft Q-learning. ArXiv, March.
>
> [4] Weighted QMIX: Expanding Monotonic Value Function Factorisation for Deep Multi-Agent Reinforcement Learning. Tabish Rashid, Gregory Farquhar, Bei Peng, Shimon Whiteson. NeurIPS 2020.
>
> [5] Mahajan, A., Rashid, T., Samvelyan, M., & Whiteson, S. (2019). MAVEN: Multi-agent variational exploration. Advances in Neural Information Processing Systems, 32(NeurIPS).
>
> [6] Wang, J., Ren, Z., Liu, T., Yu, Y., & Zhang, C. (2020). QPLEX: Duplex Dueling Multi-Agent Q-Learning. 1–16. http://arxiv.org/abs/2008.01062
>
> [7] Wang, Y., Han, B., Wang, T., Dong, H., & Zhang, C. (2020). Off-Policy Multi-Agent Decomposed Policy Gradients. Cdm, 1–20. http://arxiv.org/abs/2007.12322
>
> [8]Rashid, T., Samvelyan, M., de Witt, C. S., Farquhar, G., Foerster, J., & Whiteson, S. (2020). Weighted QMIX: Expanding Monotonic Value Function Factorisation. ArXiv.

---

### Decision · Program_Chairs · 2021-01-07
**Final Decision**

**Decision:**

Reject

**Comment:**

Although the paper presents some interesting ideas, in general the reviewers agree that the paper lacks clear results and is not an easy read. The paper proposes a factorisation of value functions, a topic that has received quite some attention in the literature (e.g. QPLEX), and it seems that their is not sufficient innovation in the proposed method in the paper. There are also a number of claims in the paper (e.g. partial observability etc.) with which some of the reviewers disagree, and should be discussed more carefully in a revised version of the article, that all in all seems to need more work.